# Nanoparticle ocular immunotherapy for herpesvirus surface eye infections evaluated in cat infection model

**Michael Lappin\*, Kathryn Wotman, Lyndah Chow, Maggie Williams, Jennifer Hawley, Steven Dow\***

From the Translational Medicine Institute, Department of Clinical Sciences, College of Veterinary Medicine and Biomedical Sciences, Colorado State University, Fort Collins, Colorado, United States of America

\* sdow@colostate.edu (SD); Michael.Lappin@colostate.edu (ML)

**Data Availability Statement:** All relevant data are within the paper.

**Funding:** These studies were supported by a grant from the Translational Medicine Institute at

## Abstract

Ocular herpes simplex type 1 (HSV-1) infections can trigger conjunctivitis, keratitis, uveitis, and occasionally retinitis, and is a major cause of blindness worldwide. The infections are lifelong and can often recrudesce during periods of stress or immune suppression. Currently HSV-1 infections of the eye are managed primarily with anti-viral eye drops, which require frequent administration, can cause irritation, and may take weeks for full resolution of symptoms. We therefore evaluated the effectiveness of an ocular immune activating nanoparticle eye drop as a novel approach to treating HSV-1 infection, using a cat feline herpesvirus -1 (FHV-1) ocular infection model. In vitro studies demonstrated significant induction of both type I and II interferon responses by the liposome-dual TLR 3/9 agonist nanoparticles, along with suppression of FHV-1 replication. In cats with naturally occurring eye infections either proven or suspected to involve FHV-1, ocular nanoparticle treated animals experienced resolution of signs within several days of treatment, including resolution of keratitis and corneal ulcers. In a cat model of recrudescent FHV-1 infection, cats treated twice daily with immune nanoparticle eye drops experienced significant lessening of ocular signs of infection and significantly fewer episodes of viral shedding compared to control cats. Treatment was well-tolerated by all cats, without signs of drug-induced ocular irritation. We concluded therefore that non-specific ocular immunotherapy offers significant promise as a novel approach to treatment of HSV-1 and FHV-1 ocular infections.

## Introduction

Herpes simplex virus-1 (HSV-1) active replication in ocular tissues and trigger a variety of diseases, including conjunctivitis, epithelial and stromal keratitis, vasculitis, and uveitis, all of which occur more commonly following reactivation of latent HSV virus infection in the trigeminal ganglion [1, 2]. Less commonly, the posterior segment of the eye may be involved in HSV-1 associated ocular inflammation. All layers of the cornea are susceptible to active HSV-1 infection and subsequent damage from the local immune response. Disease manifestations

Colorado State University and from the state of Colorado. The funders played no role in either the study design or the study performance and manuscript preparation.

**Competing interests:** I have read the journal's policy and the authors of this manuscript have the following competing interests: SD, KW and LC currently hold US patent "Compositions and methods for enhancing innate immunity in a subject for treatment of infections and cancer and other acute and chronic conditions of the eye" SD and LC and ML also hold equity in a start-up company (Laporte Immunotherapeutics) that is developing the ocular immunotherapy technology

include corneal epithelial inflammation which may present as keratitis as well as punctate or geographic corneal ulceration [3, 4]. Severe consequences of active infection include loss of corneal sensation and development of neuropathic keratopathy, as well as non-healing epithelial defects and occasionally corneal perforation [2]. Following initial HSV-1 infection of the cornea, chronic stromal keratitis may be driven in part by immune-mediated inflammation in addition to active infection of corneal epithelial cells [5, 6]. Corneal stromal disease represents one of the largest threats to vision due to the vascularization, fibrosis and thinning of the cornea that occurs with each bout of herpetic induced inflammation.

Treatment of HSV-1 ocular surface infections typically consists of frequent administration of antiviral eye drops such as trifluridine drops, which are administered as frequently as 9 times per day because of the virostatic mechanisms of action, and failure to persist in corneal tissues, in addition to relatively poor corneal penetration. Significant local toxicities may develop following treatment with nucleoside analogues due to their indiscriminate targeting of DNA replication in both normal and virally-infected cells [7, 8].

Oral drugs used to target HSV-1 ocular infections include acyclovir, valaciclovir, and famciclovir, and may be most effective at decreasing recurrence of latent HSV-1 infection [9, 10]. Systemically administered anti-viral drugs are often used in combination with topical anti-inflammatory drugs and anti-viral drugs during active disease periods. This increases the treatment burden for patients and although uncommon, HSV-1 can develop resistance to therapies such as acyclovir [7]. While effective in most cases of HSV-1 keratitis and conjunctivitis, the need for very frequent administration of anti-viral eye drops, the slow response times, and the associated ocular irritation from anti-viral eye are all issues with this form of disease management. Therefore, there remains a strong need for alternative or adjunctive approaches to the management of ocular HSV-1 infections.

As an alternative or adjunct to anti-viral eye drops or oral medications for treatment of HSV-1 ocular infections, topical immunotherapy offers several potential advantages. An immunotherapy that allows for potent activation of innate immunity and stimulation of broad interferon responses, including IFN-α, IFN-β, and IFN-γ, locally in ocular tissues can suppress herpesvirus replication as activation of type I interferons have been shown to enhance the ability of corneal epithelial cells to decrease herpesvirus entry into healthy cells [[11]; 5]. Development of resistance is also less likely to occur with non-specific immunotherapy than with antiviral drugs, where herpesvirus resistance is well-documented [7, 12, 13]. Moreover, activation of potent local ocular immune responses could also lead to the generation of systemic, T cell mediated immunity, which would in turn reduce the likelihood of reactivation of latent herpesvirus infection in trigeminal ganglia. Thus, topical ocular immunotherapy has the potential to control both acute and chronic, recurrent ocular herpesvirus infections.

In the current study, we adapted liposome-dual TLR agonist complexes (LTAC) nanoparticles that we previously evaluated for non-specific protection from respiratory tract infections to use as a topical ocular immunotherapeutic [14–16]. The LTAC nanoparticles used here consisted of cationic liposomes complexed to a TLR 3 agonist (polyinosinic, polycytidylic acid) and a TLR 9 agonist (non-coding plasmid DNA). The use of cationic liposomes significantly improves the entry of the nucleic acid agonists into endosomal pathways, where TLR 3 and TLR9 are expressed, as reported by us and others [17–19]. Importantly, introduction of double-stranded RNA and CpG oligonucleotides into the cytoplasm can also trigger non-TLR intracellular pattern recognition pathways, including RIG I and the cyclic-GMP-AMP synthase (c-GAS) pathways, resulting in activation of interferon synthesis pathways including the STING pathway [20–22]. Indeed, liposomal-plasmid DNA complexes very similar to those used here have been shown to activate the STING pathway [20]. The LTAC complexes used in these studies also contained carboxymethylcellulose (CMC), which was added to improve

adherence to epithelial cells including the conjunctival andcorneal epithelium. We previously demonstrated increased adherence of CMC-coated nanoparticles to epithelial cells in vitro, which was also accompanied by enhanced immune activation and interferon production [15]. Thus, multiple complementary, anti-viral immune pathways could be activated locally by LTAC nanoparticles when applied to mucosal surfaces.

Earlier versions of these immune-activating nanoparticles have been widely evaluated previously in multiple different animal viral and bacterial infection models for generation of non-specific protective immunity. For example, immune nanoparticles delivered by inhalation or injection were shown to trigger protection from diverse pathogens, including *Burkholderia*, *Francisella*, *Yersinia*, and phlebovirus infections [23–26]. In cats, we have shown that prophylactic treatment with LTAC delivered as an intranasal and pharyngeal spray reduced the severity of FHV-1 associated respiratory disease and decreased viral shedding after challenge infection [27]. Similar protective immunity was also demonstrated in dogs with canine herpesvirus infection and in cattle with mixed viral and bacterial pneumonia [15].

The ocular immunotherapy studies reported here utilized cat natural and experimental [28] FHV-1 infection models, which have been shown to recapitulate many of the clinical abnormalities associated with HSV-1 ocular infection in humans [29, 30]. Recurrent ocular disease in cats is a common manifestation of chronic infection with FHV-1 [31–34]. Up to 97% of the cat population becomes infected with FHV-1 by adulthood, and infection is typically lifelong [33, 34]. The ocular manifestations of chronic and recurrent FHV-1 infection in cats include conjunctivitis, keratitis, and corneal ulceration [31, 34]. Endogenous uveitis has also been associated with FHV-1 in some cats [35]. In some cases, persistent corneal injury can be severe enough to induce corneal opacity and complete loss of vision. Moreover, the cat as an outbred species with naturally occurring infections and frequent recrudescence following stress can be expected to more accurately recapitulate the natural history of human HSV infections and responses to treatment than experimental rodent models of infection.

Therefore, in the present study we investigated whether immune stimulatory LTAC nanoparticles were effective and safe as a topical ocular immunotherapy to treat surface oriented ocular herpesvirus infections in cats. We reasoned that nanoparticle eye drops could activate innate immune responses in subconjunctival tissues and the corneal epithelium and suppress herpesvirus replication and signs of infection. We report here that treatment with LTAC eye drops resulted in rapid resolution of clinical signs in cats with naturally occurring ocular FHV-1 infection, including a subset of cats refractory to conventional treatment. In experimental infection recrudescence studies, treatment with LTAC significantly lessened ocular signs of infection and lead to significantly fewer episodes of viral shedding compared to control cats. Thus, we concluded that ocular immunotherapy with LTAC nanoparticle eye drops has promise as a new approach to management of ocular surface herpesvirus infections in humans and cats.

## Materials and methods

### Nanoparticle formulation

Immune stimulatory LTAC nanoparticle eye drops used in this study were prepared as previously described [15]. Briefly, cationic liposomes comprised of DOTAP (1,2-dioleoyl-3-trimethylammonium-propane) (Avanti Polar Lipids Inc, Alabaster, AL) and cholesterol were complexed to polyinosinic, polycytidylic acid (Sigma-Aldrich, St. Louis, MO) and non-coding plasmid DNA (Aldevron, Fargo, ND), after which carboxymethylcellulose (Sigma-Aldrich) was added to the preformed complexes, as noted previously [15]. The LTAC nanoparticles were placed in sterile dropper bottles for administration as eye drops.

## Cell lines and FHV-1 stocks

Crandall-Reese feline kidney cells (CRFK) and Fcwf-4 cells (feline macrophage cell line) were purchased from ATCC (American Type Culture Collection, Manassas, VA) and maintained in DMEM (Gibco-ThermoFisher Scientific, Waltham, MA) with 10% FBS (Atlas Biologicals, Ft Collins, CO) and penicillin-streptomycin solution. Cells were screened for mycoplasma contamination prior to use in assays. Cells were seeded in triplicate or quadruplicate wells of 96-well plates (Corning Inc, Corning, NY) at a density of 5 X $10^4$ cells per well prior to inoculation with FHV-1 or treatment with LTAC. FHV-1 strain 96–13 was initially acquired from the United States Department of Agriculture (USDA).

## Viral quantification and serology

Total DNA was extracted from ocular swabs and quantitative RT-PCR was used to amplify FHV-1 and feline 28S ribosomal DNA [control] as described previously [27, 31].Serum FHV-1 IgG antibody titers were measured using an ELISA, as described previously [27].

## In vitro immune activation assays

The impact of LTAC nanoparticle treatment on induction of interferon production by feline immune cells (macrophages) was assessed using a feline macrophage cell line (Fcwf-4, American Type Tissue Collection, Manassas, VA). The Fcwf-4 cells were seeded at a density of 1 X $10^5$ cells per well in triplicate wells of 24-well plates, in complete medium, and allowed to adhere overnight as previously reported. The cells were then incubated with varying concentrations of the LTAC nanoparticles in complete medium or were left untreated. After 8 hours in culture, RNA was extracted from the cells using RNeasy kit (Qiagen, Hilden, Germany) and IFN-α and IFN-β transcript abundance was determined by qRT-PCR. RT-PCR was performed using GoTaq® 1-step-PCR-system (Promega, Madison, WI) on QuantStudio 3 Real-Time PCR system (Thermo Fisher Scientific, Waltham, MA). Primer and probe sequences are shown below. All primer and probe pairs were designed using IDT primer quest tool (Integrated DNA Technologies, Coralville, IA). (IFN-γ, IFN-α, IFN-β, RPL30 housekeeping gene) and were verified by standard curve to have efficiency >95%.

| Feline IFNa1 NM_001245020 | Fwd: CTGTCAGAAGGACAGAAGTG |
| --- | --- |
| | Rev: GTGCAGAAGAAGTGGAAGAT |
| | Probe: /56-FAM/TGTTTGGTG/ZEN/GAGACCAGTCCCACAA/3IABkFQ/ |
| Feline IFNb1 NM_001009297.1 | Fwd: CACTGTTGAGAACCTCCTTG |
| | Rev: GGTTCAGAAGGGTCGTATTG |
| | Probe: /56-FAM/TGGCAGAAG/ZEN/GAACACCTGGAAACGA/3IABkFQ/ |
| Feline RPL30 XM_006943321.4 | Fwd: AAGCGAAACTGGTCATCC |
| | Rev: ATTGCCGCTGTAGTGATG |
| | Probe: /56-FAM/AACAACTGC/ZEN/CCAGCCTTGAGGAAGT/3IABkFQ/ |

## In vitro suppression of FHV-1 replication by nanoparticles

Feline Fcwf-4 cells were seeded at a density of 1 X $10^5$ cells per well in triplicate wells of 24-well plates, in complete medium and allowed to adhere overnight. To assess the effects of LTAC nanoparticle-induced innate immune activation on suppression of FHV-1 replication, the Fcwf-4 cells were treated with LTAC nanoparticles diluted in complete medium at concentrations of 5 ul/ml, 10 ul/ml, and 25 ul/ml, for 6 hours at 37°C. The medium was then removed and replaced with complete medium or for 2 hours with medium containing FHV-1 diluted to

a titer of $10^{7.28}$/ml TCID$_{50}$, after which the inoculum was removed and cells were overlayered with a 3% agarose solution in complete medium as described previously [36]. The agarose solution was allowed to cool and gel at room temperature for 15 minutes, after which the plates were placed in a 5% CO2 incubator at 37°C and incubated for an additional 96 hours. After 96 hours, 3.7% formaldehyde was added to each well for 20 minutes and then 0.3% crystal violet in 25% methanol in water was added to each well for 10 minutes to stain cells. The crystal violet solution was then washed out with deionized water twice. The percentage cell confluency as a measure of FHV-1 induced cytopathic effect was then determined using an Incucyte instrument (Essen Biosciences, Ann Harbor, MI).

## Study design, cats with suspected or proven naturally-occurring FHV-1 ocular infection

Cats > 8 weeks of age were eligible for entry into this open clinical trial evaluating ocular nanoparticle immunotherapy study. The study cats were recruited from local animal shelters in the vicinity of the Colorado State University Veterinary Teaching Hospital (CSU-VTH) and was approved by the Institutional Animal Care and Use Committee (Protocol 19-9574A) and by the Boards of Directors of the participating shelters.

Cats with signs consistent with conjunctivitis including conjunctival hyperemia, chemosis, and ocular discharge qualified for the study provided the total ocular lesion score (sum of all ocular signs present in both eyes) was ≥ 2 and the ocular clinical signs of disease had been present for at least 2 days with no change or worsening observed during that time (**Table 1**). Cats that had failed other treatments for ocular inflammation were accepted. Many cats with suspected viral infections have both ocular and respiratory signs of disease and many cats with a primary viral infections often have secondary overgrowth of local bacterial flora (33).Thus, the protocol allowed use of oral amoxicillin for some study cats, according to the International Society for Companion Animal Infectious Diseases respiratory treatment guidelines [37]. Cats with severe corneal ulcers or corneal sequestra that required surgery and cats infected with feline leukemia virus or feline immunodeficiency virus were excluded from the study.

After initial case assessment using a direct ophthalmoscope (shelter veterinarians) or slit lamp (cases referred to Colorado State University were examined by KW), the attending veterinarian took photographs of both eyes and then gently rolled a swab in the conjunctival fornix of most cats for collection of cells for extraction of nucleic acids and performance of PCR

**Table 1. Ocular disease scoring rubrics.**

| Condition | Ocular Lesion Score | Clinical Signs |
|---|---|---|
| Conjunctivitis | 0 | none |
| | 1 | Mild conjunctival hyperemia |
| | 2 | Moderate to severe conjunctival hyperemia |
| | 3 | Moderate to severe conjunctival hyperemia and chemosis |
| Blepharospasm | 0 | none |
| | 1 | Eye < 25% closed |
| | 2 | Eye 25% - 50% closed |
| | 3 | Eye > 50% closed |
| Ocular discharge | 0 | none |
| | 1 | Serous |
| | 2 | Mucoid |
| | 3 | Mucopurulent |

assays for FHV-1, feline calicivirus, and *Mycoplasma* spp. using standardized assays. Fluorescein stain was then applied to assess for corneal epithelial disruption. Cats were treated at the shelter or transferred to be treated at the Colorado State University Veterinary Teaching Hospital (CSU-VTH) for further treatment and evaluation. A standardized clinical scoring system (Table 1) that was previously reported [34] in a study of cidofovir for the treatment of FHV-1 was used for each cat once daily, using two trained individuals masked to the treatment groups. Cats that worsened on LTAC nanoparticle treatment were removed from the study and an antibiotic administered at the discretion of the study ophthalmologist (KW), based on assessment of conjunctival cytology. For cats that failed treatment with LTAC nanoparticles, cidofovir (compounded as a 0.5% solution) was administered as eye drops twice daily. For cats that failed cidofovir treatment, famciclovir was administered (90 mg/kg, p.o., every 8 hours). After completion of the study protocol, all cats were returned to their respective shelters for adoption.

## Study design, experimental FHV-1 re-infection

Young adult (11 months of age), purpose-bred research cats (n = 18) that had been previously infected with the USDA 96–13 strain of FHV-1 and currently had FHV-1 serum neutralization titers that ranged from 1:16 to 1:64 (cats with negative FHV-1 titers were excluded) were available for study in our colony. These studies were approved by the facility IACUC (protocol 170.056). Previous FHV-1 infection does not induce sterilizing immunity and we therefore believed that mild illness would result after repeat exposure to the virus and would allow us therefore to assess a treatment effect. To trigger FHV-1 ocular re-infection and conjunctivitis, cats were administered the arbitrary dose of $10^{5.68}$ TCID50 FHV-1 in a volume of 25 µl placed into the conjunctival fornix of both eyes.

Study cats were randomly assigned to a treatment group of n = 9 cats (1 drop of LTAC nanoparticles administered in both eyes twice daily), or to a control group of n = 9 cats (1 drop of liquid tear replacement; GenTeal Tears; Alcon) administered in both eyes twice daily. A standardized clinical scoring system (Table 1) that was previously reported [34] in a study of cidofovir for the treatment of FHV-1 was used for each cat once daily, using two trained individuals masked to the treatment groups. The trained observers consisted of 4 DVM students or research associates that were trained by a study veterinarian (ML) to apply the clinical scoring system. If blepharospasm was detected during the study, the observers were advised to contact a study veterinarian (KW or ML) for a full ophthalmic examination and assessment for keratitis. Treatment was initiated when one or both eyes had an ocular score of > 1. Cats that had not achieved a score of > 1 in either eye by Day 5 were treated with their randomized product in both eyes. All study cats were treated each for a total of 10 days. Samples were collected from the conjunctival fornix of both eyes for quantitative FHV-1 PCR assay twice weekly for 21 days, starting 3 days after FHV-1 re-infection.

## Statistical analyses

For in vitro studies, comparisons amongst controls and treated cells for cytokine concentrations and FHV-1 induced cytopathy were compared using ANOVA, followed by the Tukey multiple means post-test, using Prism 8 software (GraphPad, San Diego, CA). Data for the clinical study cats are presented descriptively. Each eye was assessed individually and when the ocular score for that eye decreased $\geq$ 50% from baseline, the response was considered positive. For the experimental FHV-1 re-challenge study, the proportions of eyes with FHV-1 associated disease (any eye with a cumulative score > 1)) and the proportion of eyes with positive FHV-1 PCR assay results were calculated for the 10 day treatment period and the results were

compared statistically between groups using Fishers Exact test. Delta-delta Cq values were calculated for all samples positive for FHV-1 DNA and 28S ribosomal DNA as an estimate of viral load and results compared between groups. This data was not normally distributed and so the median values were compared by Mann Whitney U test. Statistical significance levels for all analyses were defined as $p < 0.05$.

## Results

### Activation of Type I interferon responses by LTAC nanoparticles

The composition of the LTAC nanoparticles used in the present study is illustrated in the diagram in **Fig 1**. Key features of the nanoparticles are the net positive charge imparted by the cationic liposomes, which facilitates binding and intracellular uptake by target cells for viral infection, including epithelial cells and immune cells, especially macrophages. In addition, the particles are coated with low-molecular weight CMC, which serves to further improve adherence to epithelial surfaces, as reported recently [14]. In addition, the addition of CMC to the nanoparticles increased their overall immune potency and duration of immune stimulatory activity.

In this study, we measured the ability of LTAC nanoparticles to induce production of type I interferons by feline macrophages, as these are one of the cells most likely to initially respond to immune stimulation in the subconjunctival tissues. Type I interferons (both IFN-α and IFN-β) are key antiviral cytokines known to play an important role in the innate immune control of herpesvirus infections [2, 5]. We observed that feline macrophages stimulated with LTAC produced both IFN-α and IFN-β transcripts at high levels (**Fig 2**), consistent with the

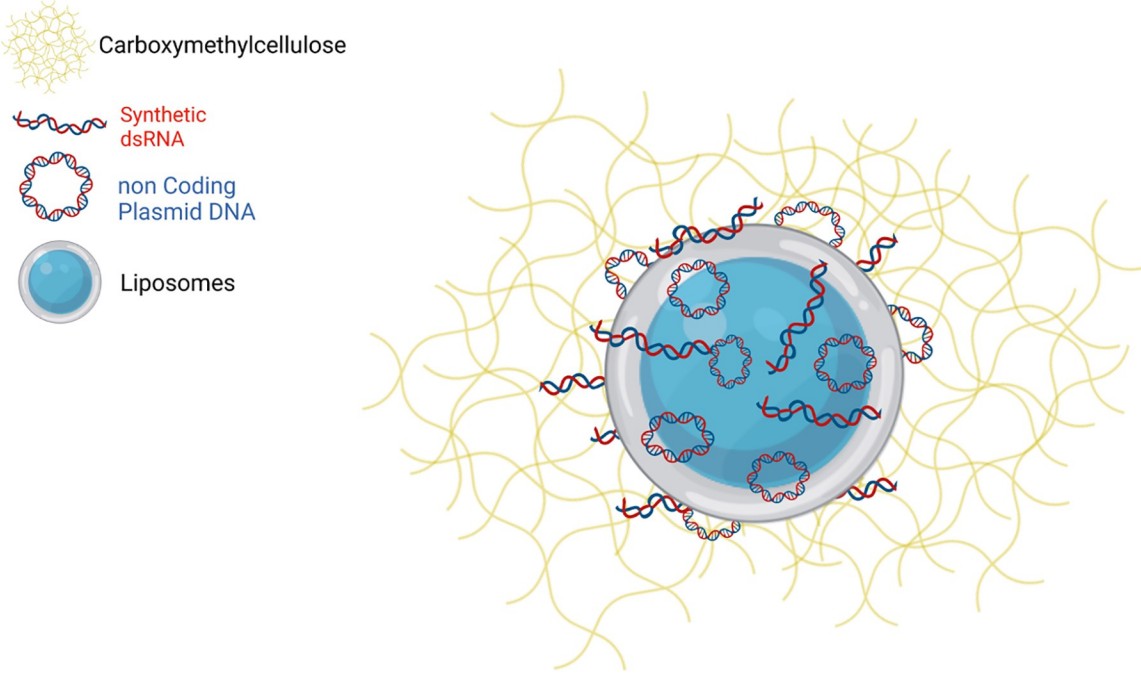

**Fig 1. Diagram of LTAC immune nanoparticles.** The LTAC nanoparticles consist of a cationic liposome core, to the surface of which immune stimulatory nucleic acids polyinosinic, polycytidylic acid (synthetic dsRNA), which is a TLR3 agonist; and CpG ODN, (non coding plasmid DNA) which is a TLR9 agonist are complexed by charge interactions. The addition of CMC (Carboxymethylcellulose) after the liposome-nucleic acid complexes are formed provides a loose outer coating to increase adherence to the surface of mucosal tissues, including epithelial cells, and improve overall uptake of the LTAC complexes.

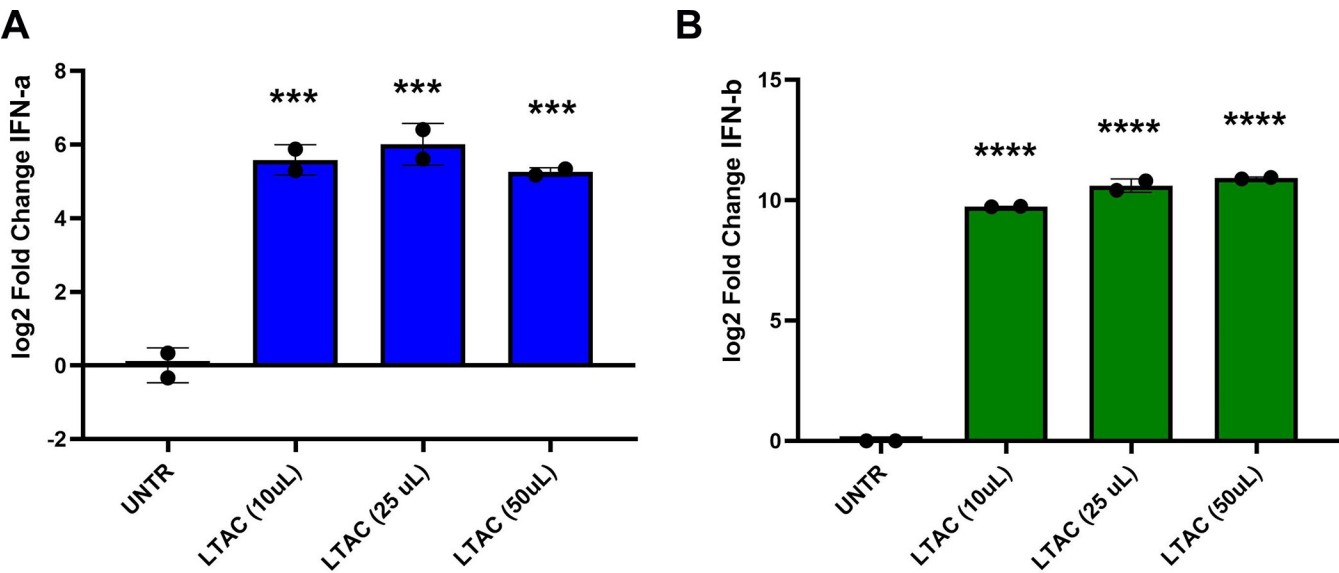

**Fig 2. Activation of type I interferon responses by in vitro treatment with LTAC nanoparticles.** Feline Fcwf-4 macrophages were seeded in 24-well plates overnight, and then stimulated for 8 hours with 3 different concentrations of LTAC nanoparticles. Following treatment, RNA was extracted and transcript abundance for **A**) interferon-α and **B**) interferon-β determined by quantitative RT-PCR. Significant induction of both IFN-α and IFN-β transcripts was observed at all 3 LTAC concentrations evaluated. Statistical evaluation of study data was done using ANOVA, and Tukey-Kramer post-test, with significance set at $p < 0.05$. Similar results were obtained in one additional, independent experiment.

previously reported ability of liposome-TLR complexes to stimulate type I IFN production in other species [17]. Thus, the LTAC nanoparticles were readily internalized by feline macrophages, and rapidly stimulated production of type I interferons [14].

## Suppression of FHV-1 replication in vitro by LTAC nanoparticles

Next, we investigated whether LTAC nanoparticle activation of feline macrophages could suppress FHV-1 replication *in vitro*. To address this question, Fcwf-4 feline macrophages were pre-treated for 6h with LTAC at varying concentrations, after which the cells were inoculated with FHV-1, and the effect on FHV-1 induced cytopathic effects was assessed (**Fig 3**). These experiments demonstrated that cells activated with LTAC nanoparticles suppressed the cytopathic effects of FHV-1 on macrophages, consistent with suppression of viral replication in the cells. These findings are important because macrophages are one of the target cells for herpesvirus infection and replication at mucosal sites [5]. Thus, LTAC nanoparticles exerted both type I interferon stimulating properties and anti-viral activity against FHV-1.

## Responses to ocular immunotherapy in cats with suspected naturally occurring FHV-1 ocular infections

This study was designed to assess the impact of LTAC immunotherapy on disease manifestations in cats with naturally occurring ocular disease that is likely to related to FHV-1 infection. Each of the 10 cats evaluated in this experiment were considered by the primary veterinarian to have typical signs of ocular FHV-1 (**Fig 4**). Positive FHV-1 PCR assay results confirmed the infection in all 8 eyes with samples available for testing. A total of 13 eyes were affected and the cases are summarized in **Table 2**. Cats were treated twice daily with 1 drop of LTAC nanoparticles administered to each affected eye. Cats were evaluated daily by a trained observer at each treatment site, and ocular scores were recorded daily for 7 days of observation, as described in

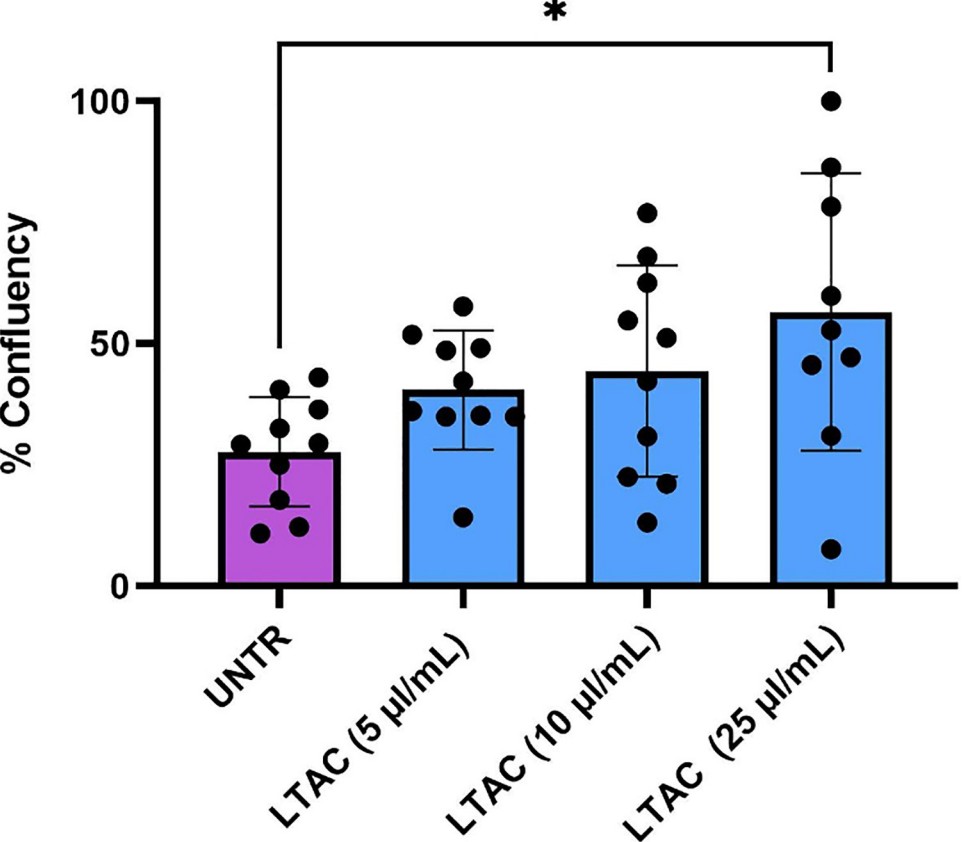

**Fig 3. Suppression of FHV-1 induced cytopathic effects by pre-treatment with LTAC nanoparticles.** Feline Fcwf-4 macrophages in triplicate wells were treated with 3 different concentrations of LTAC nanoparticles for 6 hours, after which the nanoparticles were removed and the cells infected with FHV-1 for 2 hours as described in Methods. Cell confluency as a measure of cytopathic effects was assessed by Incucyte at 48 hours for untreated, uninfected cells, FHV-1 infected and untreated cells, and for FHV-1 infected and LTAC-treated cells. Statistical evaluation of study data was done using ANOVA, and Tukey-Kramer post-test, with significance set at $p < 0.05$. Similar results were obtained in two additional, independent experiments.

Methods. For 11 of the 13 treated eyes, a positive response (partial or complete resolution of ocular inflammation and lesions) was observed by day 7 of treatment. The median, mean, and ranges for time to treatment responses were 1 day, 2.18 days (s.d. = 1.7 days), and 1–5 days,

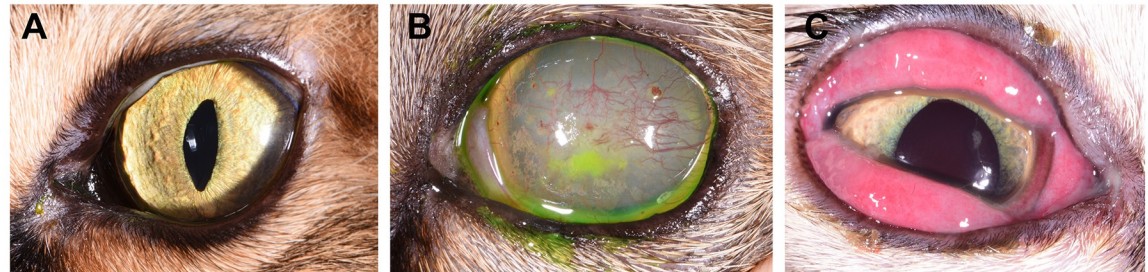

**Fig 4. Typical lesions associated with ocular infection with the feline herpesvirus FHV-1 in naturally infected cats.** Photographs depict signs of infection in naturally infected cats: A) healthy cat eye; B) severe corneal inflammation and resulting keratitis, and C) conjunctivitis with significant hyperemia and swelling.

**Table 2. Clinical responses of cats with naturally occurring conjunctivitis/keratitis when administered the LTC nanoparticle therapy.**

| Cat # | Prior Treatment | Eye | PCR Result | Ocular Score Day of Therapy | | | | | | | | First Day of Positive Response |
|---|---|---|---|---|---|---|---|---|---|---|---|---|
| | | | | Day 0 | Day 1 | Day 2 | Day 3 | Day 4 | Day 5 | Day 6 | Day 7 | |
| 1 | Unknown | OS | NS | 4 | 1 | 1 | 1 | 0 | 0 | 0 | 0 | 1 |
| 2 | Unknown | OS | FHV-1 | 8 | 3 | 2 | 3 | 3 | NS | 2 | 2 | 1 |
| 3 | Unknown | OS | FHV-1 | 2 | 0 | 0 | 0 | 0 | NS | 0 | 0 | 1 |
| 4 | Unknown | OD | FHV-1 | 3 | 1 | 1 | 1 | 1 | NS | 1 | 1 | 1 |
| 5 | Unknown | OD | FHV-1, Myco | 9 | 6 | 2 | 2 | 1 | NS | 1 | 1 | 2 |
| 6* | Unknown | OS | NS | 2 | 3 | 4 | 2 | 2 | 4 | 4 | 2 | NA |
| 7 | None | OD | NS | 6 | 2 | 1 | 0 | 1 | 0 | 0 | 0 | 1 |
| 8** | Failed clavamox (unknown dose), famcyclovir 90 mg/ Kg PO BID× 21 davs. doxvcycline (unknown dose): topical duramorph 0.1%/ Optix/Erythromycin/tobr amycin; simbadol prior to LTAC administration | OS | NS | 4 | 0 | 0 | 1 | 1 | 0 | 1 | 1 | 1 |
| | | OD | NS | 7 | 4 | 4 | 6 | 5 | 3 | 3 | 3 | 5 |
| 9*** | Failed clavamox (unknown dose), famcyclovir 90 mg/ Kg PO BID x 21 days, doxycycline (unknown dose): topical duramorph 0.1% Optix/Erythromycin/tobr amycin; simbadol prior to LTAC administration | OS | FHV-1 | 4 | 4 | 5 | 4 | 2 | NS | NS | 1 | 4 |
| | | OD | FHV-1 | 1 | 1 | 0 | 0 | 0 | NS | NS | 0 | 2 |
| 10**** | Failed clavamox (unknown dose), famcyclovir 90 mg/ Kg PO BID x 21 days, doxycycline (unknown dose); topical duramorph 0.1% Optix/Erythromycin/tobr amycin; simbadol prior to LTAC administration | OS | FHV-1, Myco, FCV | 6 | 6 | 5 | 5 | 5 | 5 | 6 | 6 | NA |
| | | OD | FHV-1, Myco, FCV | 6 | 6 | 5 | 5 | 4 | 3 | 3 | 3 | 5 |

*Superficial ulcer on 20% of ventral cornea on Day 0; resolving on Day 7 and was adopted

**Topical erythromycin added ODbe cause a conjunctival ulcer was noted OD

****Superficial ulcer OS on Day 0; resolving on Day 4 and adopted

****Lost to follow up after Day 7

OS = left eve: OD = right eve: FOV = feline calicivirus: Myco = Mycoplasma species

respectively (**Table 2**). In the 3 cats with bilateral ocular disease that had failed other bacterial and viral treatments, 5 of 6 affected eyes had a positive response to LTAC nanoparticles by Day 5 (**Table 2**). In the third cat with bilateral disease (positive for FHV-1, calicivirus, and mycoplasma), one eye did not respond to treatment by Day 7 (cat #10, left eye). (The cat was lost to follow-up after day 7).

One cat (cat #8, right eye) had a delayed positive response to ocular LTAC treatment, with a partial response only noted by day 4. Examination on day 4 revealed a conjunctival ulcer, and bacterial conjunctivitis and ulcer infection was suspected. This cat was administered erythromycin ocular ointment starting on Day 4 and the affected eye had a partial response to combined therapy by Day 5. Both cats (cat #6, left eye, and cat #9, left eye) with superficial corneal ulcers responded to treatment by Day 7 and Day 4, respectively. Photographic evidence of treatment responses in 3 affected cats is depicted in **Fig 5**.

## Evaluation of LTAC ocular nanoparticle immunotherapy in experimental model of recrudescent herpesvirus conjunctivitis

This repeat corneal FHV-1 inoculation experiment in purpose bred cats previously infected with FHV-1 was used next to model recrudescent herpesvirus ocular infection and to assess the impact of LTAC immunotherapy. One cat that was randomized to the LTAC group was

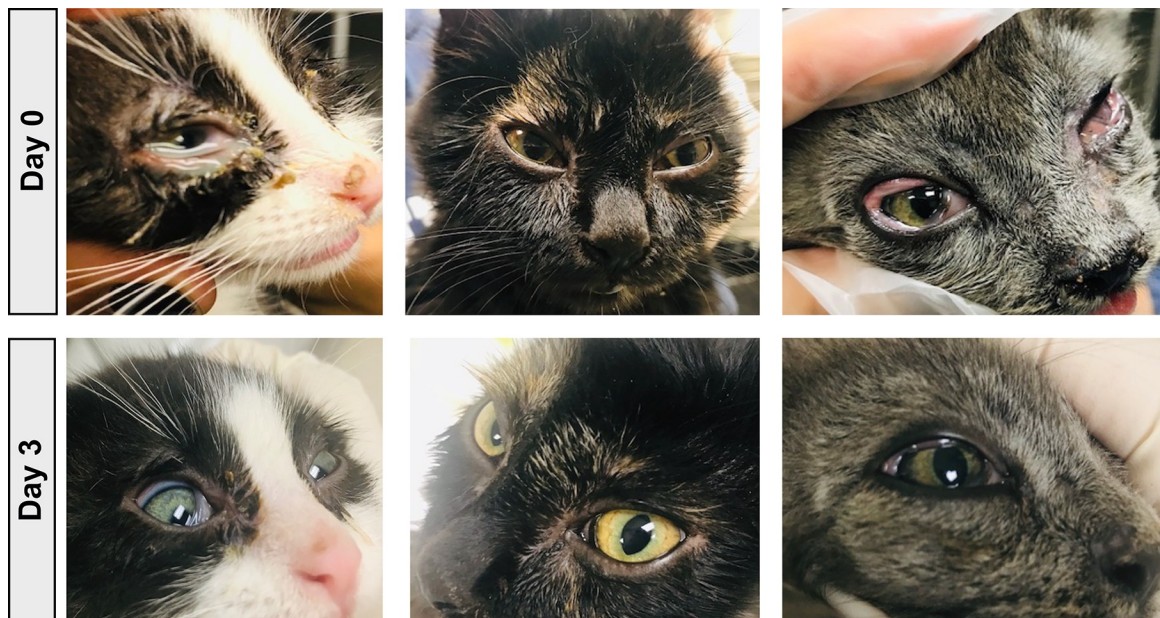

**Fig 5. Response to treatment in 3 cats with suspected naturally occurring FHV-1 surface ocular infections.** Cats with naturally-occurring surface ocular infections (conjunctivitis, keratitis) with FHV-1 and in some cases multiple pathogens were treated twice daily with LTAC nanoparticle eye drops (1 drop per eye), as noted in Methods, using an eye dropper to dispense the nanoparticles in solution. For 11 of the 13 treated eyes (Table 2), a positive response (partial or complete resolution of ocular inflammation and lesions) was observed by day 7 of treatment. The median, mean, and ranges for time to treatment responses were 1 day, 2.18 days (s.d. = 1.7 days), and 1–5 days, respectively. Only one eye failed to respond by Day 7 and that eye was positive by PCR for FHV-1, FCV, and *Mycoplasma* spp.

intractable to treatment and was removed on Day 5 of the study. None of the cats in either group required rescue therapy and both ocular preparations were well tolerated. Respiratory signs of disease were not noted in any cat.

In this model, repeat FHV-1 inoculation in previously infected cats led to induction of ocular signs of FHV-1 in one or both eyes of 10 of 17 cats that completed the experiment. As described in the methods, treatment with LTAC nanoparticle eye drops (or the placebo) was initiated in a cat on the first day inflammation was noted in either eye or on Day 5. After treatment was started, ocular signs of FHV-1 were resolved in all cats by Day 10. The proportion of inflamed eyes after treatment was started was significantly lower ($p = 0.002$) for the LTC group (13%) than the placebo group (21%) (Fig 6).

In addition, we also assessed the impact of LTAC nanoparticle eye drops (or the placebo) on ocular shedding of FHV-1. The proportion of FHV-1 positive eyes after treatment was started was significantly ($p = 0.005$) lower for the LTAC group (30%) than the placebo group (55.1%) (Fig 7). Treatment was equally well-tolerated in the LTAC treated group and the placebo-treated groups of animals. In addition, the FHV-1 and 28S ribosomal DNA Delta-Delta Cq values used to estimate viral load was numerically lower for cats administered the LTAC nanoparticles (Median = 0.14) compared to cats administered the placebo (Median = 0.06).

## Discussion

New more rapidly effective and well-tolerated treatment approaches to ocular HSV-1 infections are needed, either as stand-alone therapies or as adjuncts to anti-viral drugs, to speed virus elimination and corneal healing. In the current study, a feline model of ocular FHV-1 which provides a realistic assessment of herpesvirus corneal and conjunctival infection in the

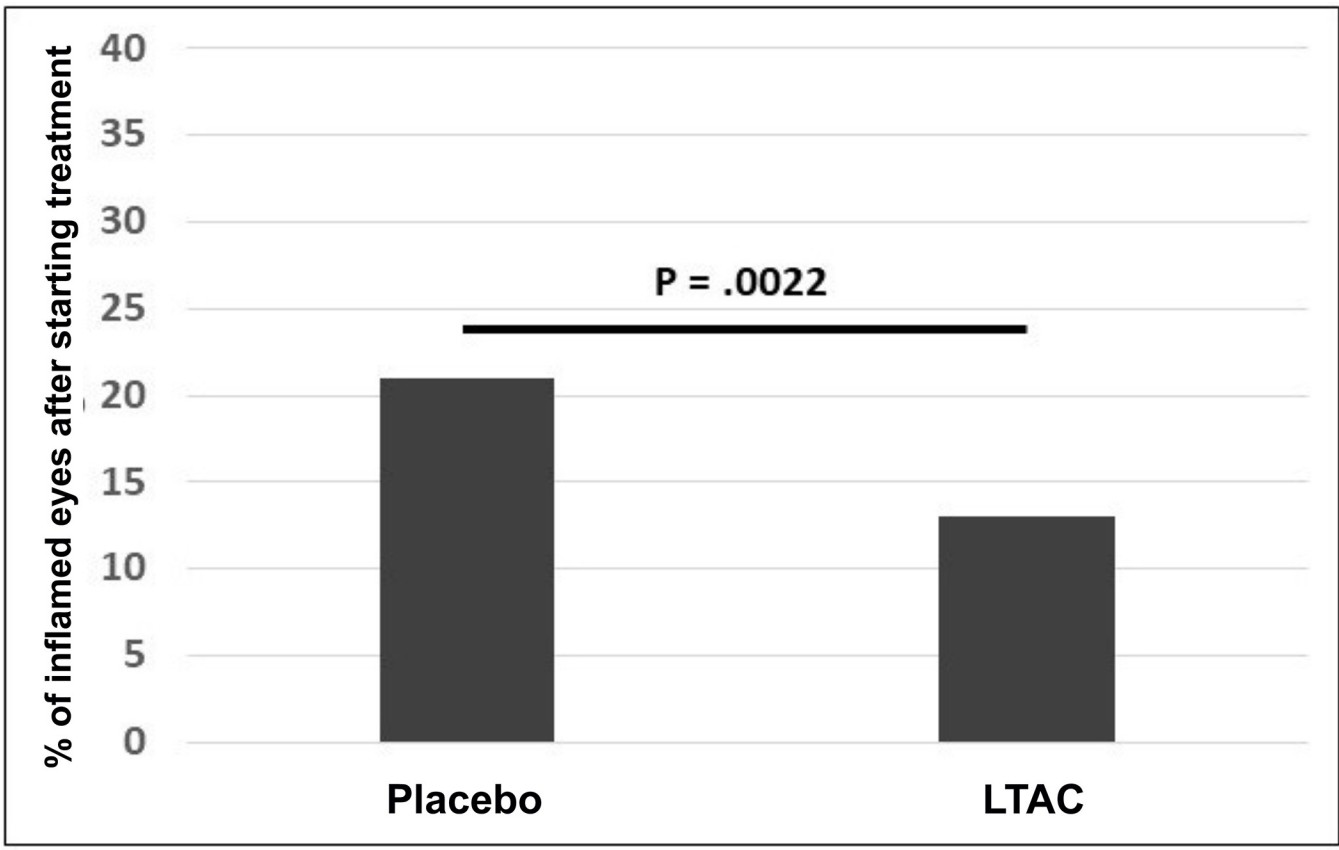

**Fig 6. Treatment responses to LTAC nanoparticle eye drops in cats with experimentally-induced recurrent FHV-1 surface ocular infection.** Purpose-bred cats previously infected with FHV-1 (at least 6 months prior) were re-inoculated by corneal instillation of FHV-1, to induce ocular infection and clinical signs. Cats were randomly assigned to an LTAC treatment group (n = 8) and a control group treated with tear replacement (n = 9) as a placebo. Both eyes were treated the first day inflammation was noted in either eye and continued for 10 days. Cats with no ocular inflammation by Day 5 were treated with the respective product for 10 days. Figure shows the proportions of inflamed eyes over the 10-day treatment period compared by Fisher's exact test and cats treated with LTAC nanoparticles experienced significantly lower days with inflammation than control cats (p = 0.002).

virus's natural host, was used to evaluate the potential impact of a novel topical ocular immunotherapy. Key finding from these investigations were that LTAC nanoparticle eye drops activated innate immune responses and type I interferon production, generated rapid clearing of signs of conjunctivitis and superficial ocular inflammation in both naturally-infected and experimentally infected cats, and lessened episodes of viral shedding in experimentally infected cats.

There are several previous reports of the ocular application of CpG oligonucleotides for evaluation of ocular immune responses, but there are to our knowledge no reports on the use to treat ocular herpesvirus infections. While generally well-tolerated by healthy eyes, it should be noted that ocular adverse effects from repeated ocular delivery of CpG oligonucleotides, including retinal and corneal inflammation, have been previously reported [38–40]. In the feline studies described herein, no LTAC nanoparticle associated inflammation was noted, but the treatment was limited to 1–2 weeks. In addition, rodent and dog 14-day LTAC treatment studies done by our group have not revealed evidence of corneal or anterior chamber inflammation (KW, unpublished data). The use of LTAC nanoparticles would likely also lead to more rapid suppression of herpesvirus replication and reduction in clinical signs of infection when co-administered with anti-viral eye drops.

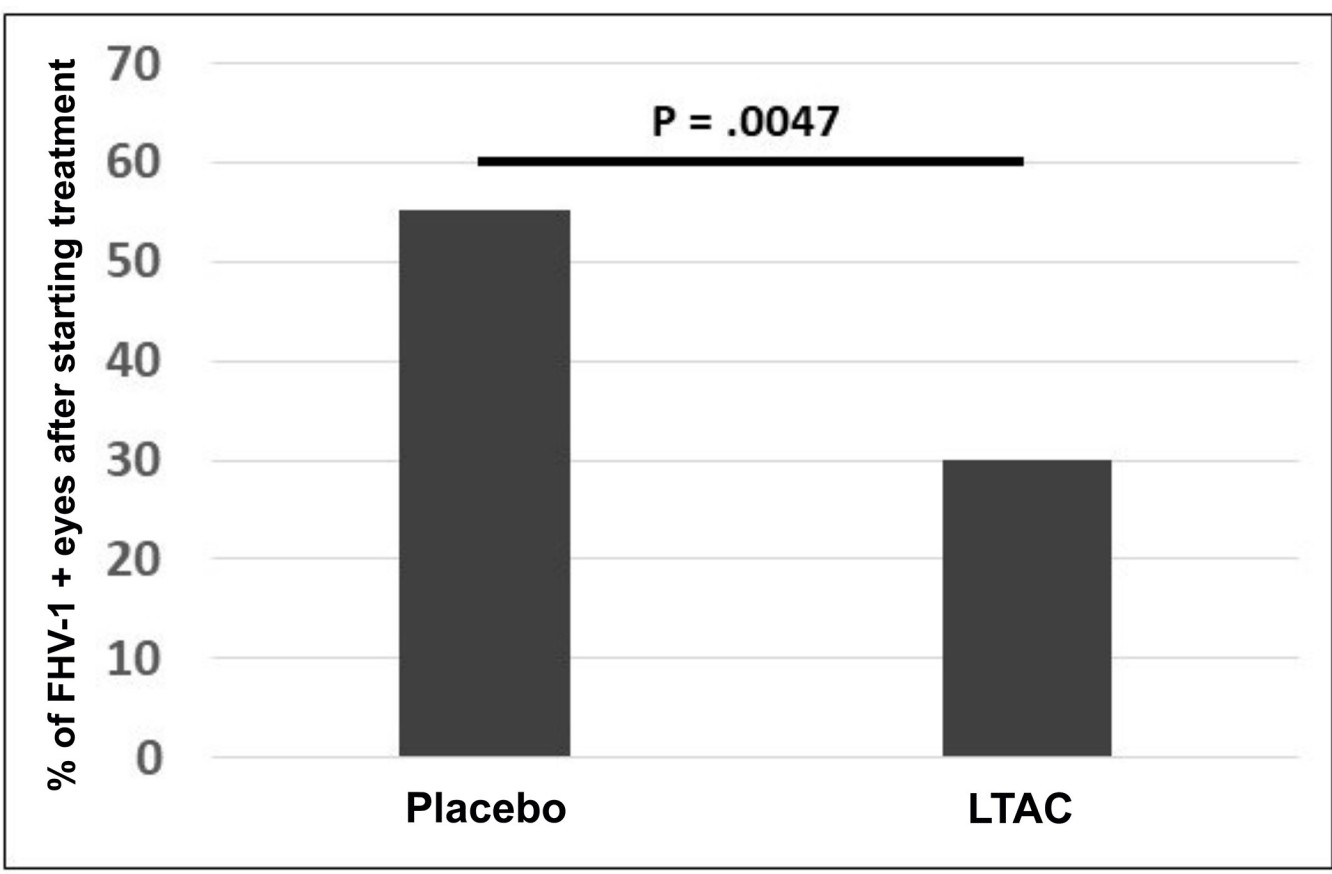

**Fig 7. Impact of LTAC nanoparticle eye drops on viral shedding in cats with experimentally-induced recurrent FHV-1 surface ocular infection.** Purpose-bred cats previously infected with FHV-1 (at least 6 months prior) were re-inoculated by corneal instillation of FHV-1. Figure shows the proportions of FHV-1 positive eyes (PCR assay) over the 10-day treatment period compared by Fisher's exact test and cats treated with LTAC nanoparticles experienced significantly lower days with inflammation than control cats (p = 0.005).

These models of ocular herpesvirus infection, using FHV-1 infected cats, resembles the HSV-1 infection in several important respects. First, FHV-1 infection of cat eyes triggers many of the same signs observed in HSV-1 infection of humans, including conjunctivitis, epithelial and stromal keratitis, and uveitis [29, 30, 33]. The cat is also the natural host for FHV-1, which can readily establish latent infections within the trigeminal ganglia. Following stress and immune suppression, FHV-1 replication can resume in corneal and periocular epithelial tissues, leading to keratitis and other ocular inflammatory responses. Immune responses to FHV-1 infection in cats are also likely to much more closely mirror those observed with HSV-1 infections in humans, given the fact that cats are an outbred species, and that their immune system has been educated by life-long FHV-1 infection. Thus, immune and antiviral responses to ocular immunotherapy targeting FHV-1 in cats are also much more likely to predict responses in humans infected with HSV-1, including the impact on recrudescent infections. It is also important to note that corneal ulcers were present in 2 cats with naturally occurring FHV-1 ocular disease and both animals had positive responses to LTAC nanoparticle therapy within the 7-day study period. Thus, the topical administration of LTAC nanoparticles appears not to interfere with corneal healing in this species.

Some cases of superficial ocular herpesvirus infections may be complicated by the presence of other pathogens, including bacteria such as *Mycoplasma felis* and *Chlamydia felis* as well as

other viruses such as caliciviruses in cats. Thus, induction of non-pathogen specific innate immune responses locally would be expected to also exert benefit against diverse other pathogens in addition to FHV-1 or HSV-1, and thereby also help non-specifically resolve mixed bacterial and viral corneal and conjunctival infections. In the experiment assessing cats with naturally occurring conjunctivitis, the one eye with FHV-1 and *Mycoplasma* spp. coinfection and one of two eyes with evidence of FHV-1, FCV, and *Mycoplasma* spp. had positive responses to LTAC nanoparticle therapy within the 7 day study period, supporting the hypothesis of a broad treatment effect against both viral and bacterial agents of surface ocular infections.

An additional advantage of ocular immunotherapy for herpesvirus infections is the potential to generate greater local and systemic cellular immunity to herpesvirus. For example, by generating strong innate immune responses locally at the ocular surface during periods when herpesviruses are actively replicating in corneal epithelial and periocular tissues, there is the potential to generate long-lasting T cell responses against the virus by activating local dendritic cells that have taken up and presented herpesvirus antigens. Such an effect has been reported previously for antigens delivered topically to the cornea [41]. Indeed, we have previously observed stimulation of greater systemic IgA responses to FHV-1 following treatment with LTAC eye drops for active ocular infection, compared to untreated animals with ocular infection. Thus, by generating herpesvirus specific T cell responses, there is the potential to suppress or eliminate episodes of viral recrudescence in the future.

There were several limitations to the study reported here which should be addressed. First, numbers of study animals, in both the natural infection model and in the experimental viral re-infection model were small and reflect the pilot nature of these investigations. In addition, in the natural or experimental FHV-1 ocular infection experiments described here, a control group of animals treated only with antiviral eye drops (eg, cidofovir drops) was not included which might have allowed for a direct comparison with ocular immunotherapy and reach definitive conclusions regarding time to resolution of clinical signs. Nonetheless, based on historical precedent and published response times to antiviral therapy in cats with FHV-1 induced ocular disease (typically 10–14 days for complete lesion resolution, versus 3–4 days reported here for LTAC-treated cats), we believe more rapid lesion and clinical resolution could be expected with LTAC nanoparticle eye drops [34, 36, 37]. Finally, herpesvirus specific cellular immunity was not addressed in this study and would be an important focus of investigation for future studies.

In conclusion, the use of non-specific local activation of innate immune responses to generate non-specific innate immune suppression of viral replication in superficial ocular tissues, holds considerable promise as a new approach to management both HSV-1 and FHV-1 ocular infections. The LTAC nanoparticles evaluated here could also be readily used in conjunction with anti-viral eye drops to generate synergistic activity against herpesvirus replication to hasten clearance and clinical improvement, while at the same time generating long-lasting HSV-1 or FHV-1 cellular immunity.

## Acknowledgments

The authors wish to thank staff at the Larimer County Humane Society and the Boulder Humane Society for their help with animal treatment and monitoring for the clinical evaluation of LTAC treatment of feline viral conjunctivitis cases.

## Author Contributions

**Conceptualization:** Michael Lappin, Kathryn Wotman, Steven Dow.

**Data curation:** Michael Lappin, Kathryn Wotman, Lyndah Chow, Maggie Williams, Jennifer Hawley, Steven Dow.

**Formal analysis:** Michael Lappin, Kathryn Wotman, Lyndah Chow, Jennifer Hawley, Steven Dow.

**Funding acquisition:** Michael Lappin, Kathryn Wotman, Steven Dow.

**Investigation:** Michael Lappin, Kathryn Wotman, Jennifer Hawley, Steven Dow.

**Methodology:** Michael Lappin, Kathryn Wotman, Steven Dow.

**Project administration:** Michael Lappin, Kathryn Wotman, Steven Dow.

**Resources:** Michael Lappin, Steven Dow.

**Software:** Lyndah Chow, Steven Dow.

**Supervision:** Michael Lappin, Kathryn Wotman, Steven Dow.

**Validation:** Michael Lappin, Kathryn Wotman, Steven Dow.

**Visualization:** Kathryn Wotman, Steven Dow.

**Writing – original draft:** Michael Lappin, Kathryn Wotman, Lyndah Chow, Steven Dow.

**Writing – review & editing:** Michael Lappin, Kathryn Wotman, Lyndah Chow, Steven Dow.

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
