## [Decision Letter · Decision Letter 0]

19 Oct 2022

PONE-D-22-23287Nanoparticle Ocular Immunotherapy For Herpesvirus Surface Eye Infections Evaluated in Cat Infection ModelsPLOS ONE

Dear Dr. Dow

Thank you for submitting your manuscript to PLOS ONE. After careful consideration, we feel that it has merit but does not fully meet PLOS ONE’s publication criteria as it currently stands. Therefore, we invite you to submit a revised version of the manuscript that addresses the points raised during the review process.

We look forward to receiving your revised manuscript.

Kind regards,

Homayon Ghiasi, PhD

Academic Editor

PLOS ONE

Journal Requirements:

"These studies were supported by a grant from the Translational Medicine Institute at Colorado State University and from the state of Colorado.  The authors wish to thank staff at the Larimer County Humane Society and the Boulder Humane Society for their help with animal treatment and monitoring for the clinical evaluation of LTAC treatment of feline viral conjunctivitis cases."

"this work was funded by Translational Medicine Institute and CSU Ventures at Colorado State University. Awarded to SD, ML and KW"

3. We note that you have a patent relating to material pertinent to this article. Please provide an amended statement of Competing Interests to declare this patent (with details including name and number), along with any other relevant declarations relating to employment, consultancy, patents, products in development or modified products etc. Please confirm that this does not alter your adherence to all PLOS ONE policies on sharing data and materials, as detailed online in our guide for authors http://journals.plos.org/plosone/s/competing-interests by including the following statement: "This does not alter our adherence to  PLOS ONE policies on sharing data and materials.” If there are restrictions on sharing of data and/or materials, please state these. Please note that we cannot proceed with consideration of your article until this information has been declared.

Reviewers' comments:

Reviewer's Responses to Questions

**Comments to the Author**

1. Is the manuscript technically sound, and do the data support the conclusions?

Reviewer #1: Yes

Reviewer #2: Partly

Reviewer #3: Yes

2. Has the statistical analysis been performed appropriately and rigorously? 

Reviewer #1: N/A

Reviewer #2: Yes

Reviewer #3: Yes

3. Have the authors made all data underlying the findings in their manuscript fully available?

Reviewer #1: Yes

Reviewer #2: Yes

Reviewer #3: Yes

4. Is the manuscript presented in an intelligible fashion and written in standard English?

Reviewer #1: Yes

Reviewer #2: Yes

Reviewer #3: Yes

5. Review Comments to the Author

Reviewer #1: This study provided sufficient experimental evidence to support the conclusion.

Overall the results are solid and consistent. Several issues still need to be addressed before publication. The following questions may help the authors to improve the quality of their already outstanding manuscript.

Reviewer #2: Check entire manuscript for incorrect use of “FVH-1” abbreviation

Check entire manuscript for typographical errors (there are many)

Introduction

In general, the authors do not adequately separate for the reader when they are referring to ocular disease associated with active, lytic HSV-1 infection vs immune-mediated disease (typically not associated with lytic infection and not treatable with conventional antivirals).

"neurotrophic keratopathy" is misspelled.

It is unclear what “latently infected corneal epithelial cells” is describing. HSV-1 does not induce latent infections in epithelial cells of any type and the presence of corneal latency is at best controversial.

“poor corneal drug penetration” is not the only or even the main reason antivirals must be administered so frequently. There virostatic nature and failure (in some cases) to persist in tissues is as important.

In clinical medicine, oral antivirals are also frequently used to treat active cytolytic HSV-1 infections.

“ocular HSV-1 ophthalmic infections” is a redundant phrase

Clearly, the statement “potent activation of innate immunity and stimulation of broad interferon responses, including IFNa, IFN-b, and IFN-g, locally in ocular and periocular tissues can suppress herpesvirus replication and do so potentially more quickly than antiviral medications” requires one or more references to support it. This entire paragraph needs more references to support the multiple, strong statements the authors are making.

The authors appear potentially confused about ocular anatomy. What are the epithelial cells in the sclera (which is located under the conjunctiva) to which they refer?

Materials and Methods

For shelter cats with naturally-acquired infections:

A detailed description of how ocular examination was performed is not provided. What instruments were used? If just gross visual examination, this is major limitation that must be acknowledged.

Conjunctival hyperemia and chemosis are typical clinical signs associated with conjunctivitis. It is not clear why the authors list them as separate inclusion criteria for the study.

The statement “mild fluorescein stain uptake consistent with mild keratitis” does not make sense. Corneal fluorescein retention indicates absence of the epithelial (with stromal exposure). This can occur with or without corneal inflammation (ie, keratitis).

Please provide a reference to support the statement “many cats with a primary viral conjunctivitis often have secondary overgrowth of local bacterial flora like Pasteurella multocida”

Fluorescein does not assess for keratitis

Can the ocular disease scoring system be referenced? This is an overly simple system that does not make much sense to this reviewer. Why was it selected when so many published, more robust systems are avaialble?

Why were some cats not sampled for PCR?

For cats with experimental infections:

-Please provide the specific age of the research cats. Depending on the FHV-1 stain used, clinical disease can be heavily influenced by the cats age.

-Why did the authors use a re-infection model? These seems like a very odd choice and is not representative of a natural scenario. Recurrent disease can be induced in latently infected cats after experimental infection using a variety of mechanisms, or you could simply have evaluated cats with primary infections. How does this system affect the ocular disease and viral shedding patterns that are observed? Please reference this model system that you state was used “to model recrudescent herpesvirus ocular infection”

-Who were the “Two trained individuals masked to the treatment groups”? Where these clinically trained ophthalmologists (veterinary or human)? Technicians? What examination equipment did they use?

Results

As before, please explain the “trained observer” in detail and how they examined the cats (including what equipment).

Discussion

“describer” typo

Another limitation is that you don’t know you were actually treating cats with FHV-1 in the shelter population (a positive PCR test can be incidental to the ocular disease).

Reviewer #3: The authors have proposed nanoparticle-based treatment for FHV-1 in cats. This model can be a substitute for the rodent models due to the scope in studying FHV-1 reactivation. The authors show that treatment with Liposome dual TLR agonist (LTAC) reduces the disease severity in naturally infected cats and in experimental viral re-infection model. Though the results presented in this work is interesting there are few concerns which needs to be addressed.

1. Fig:2- In the in-vitro study, number of biological replicates should be >3.

2. Stability and half-life of LTAC has not been estimated.

3. Fig:3 -Authors show the cytopathic effect seen in Fcwf-4 cells after FHV-1 infection. However, the actual reduction in replicating virus was not calculated. Therefore, the results do not support the authors’ statement “suppression of FHV-1 replication”. FHV-1 replication can be measured from the culture supernatant or from the cell lysate.

4. Fig:6 -The parameters used to calculate the % of inflamed eyes is not clear. Was it based on scoring? Representative eye images can help in understanding the reduction in inflammation.

5. Since FHV-1 can establish latency, the local increase in IFNs response after LTAC treatment can increase FHV-1 latency. The experiments conducted in this work were for 1-2 weeks. Is there data available for recurrence of infection in cats which were naturally infected with FHV-1?

6. PLOS authors have the option to publish the peer review history of their article (what does this mean?). If published, this will include your full peer review and any attached files.

Reviewer #1: No

Reviewer #2: No

Reviewer #3: No

---

## [Author Response · Author response to Decision Letter 0]

2 Dec 2022

Reviewer #1: This study provided sufficient experimental evidence to support the conclusion.

Overall the results are solid and consistent. Several issues still need to be addressed before publication. The following questions may help the authors to improve the quality of their already outstanding manuscript.

Reviewer #2: Check entire manuscript for incorrect use of “FVH-1” abbreviation

Check entire manuscript for typographical errors (there are many)

- Reply: The authors thank the reviewers for their efforts in reading the manuscript, catching inadvertent errors, and making helpful suggestions. We have corrected the typographical errors we could locate.

Introduction

In general, the authors do not adequately separate for the reader when they are referring to ocular disease associated with active, lytic HSV-1 infection vs immune-mediated disease (typically not associated with lytic infection and not treatable with conventional antivirals).

Reply: Thank you pointing this out, and we have attempted in the Introduction to make it clear that non-specific generation of interferon responses can suppress active, lytic viral replication, though the impact on immune mediated keratopathy is less well studied. We believe based on the observed response to antiviral immunotherapy, that the FHV-1 infection is most likely still active, albeit at a lower level than with first time infection, and thus responsive to the interferons induced by the ocular immunotherapy drops. 

"neurotrophic keratopathy" is misspelled.

It is unclear what “latently infected corneal epithelial cells” is describing. HSV-1 does not induce latent infections in epithelial cells of any type and the presence of corneal latency is at best controversial.

-Reply: The misspelling is corrected. The reference to latently infected corneal cells has now been corrected to instead refer to actively infected corneal cells (lines 56 and 59).

“poor corneal drug penetration” is not the only or even the main reason antivirals must be administered so frequently. There virostatic nature and failure (in some cases) to persist in tissues is as important.

Reply: Corrected and further noted, lines 65-66

In clinical medicine, oral antivirals are also frequently used to treat active cytolytic HSV-1 infections.

Reply: The use of oral drugs to treat active HSV-1 infections is noted in lines 69-74

“ocular HSV-1 ophthalmic infections” is a redundant phrase

Reply: Corrected, line 77

Clearly, the statement “potent activation of innate immunity and stimulation of broad interferon responses, including IFNa, IFN-b, and IFN-g, locally in ocular and periocular tissues can suppress herpesvirus replication and do so potentially more quickly than antiviral medications” requires one or more references to support it. This entire paragraph needs more references to support the multiple, strong statements the authors are making.

Reply: We have reworded this statement to reflect a current lack of head to head studies (antivirals vs immunotherapeutics) and added new references for antiviral activity of interferons for corneal herpesvirus infection. 

The authors appear potentially confused about ocular anatomy. What are the epithelial cells in the sclera (which is located under the conjunctiva) to which they refer?

Reply: Apologies for the error, this is corrected in the manuscript. It should have stated more clearly epithelial cells in the cornea and conjunctiva.

Materials and Methods

For shelter cats with naturally-acquired infections:

A detailed description of how ocular examination was performed is not provided. What instruments were used? If just gross visual examination, this is major limitation that must be acknowledged.

Reply: We have now added a description of how the shelter animals were evaluated, by either the shelter veterinarian, or by the CSU ophthalmologist (KW). Lines 217-219

Conjunctival hyperemia and chemosis are typical clinical signs associated with conjunctivitis. It is not clear why the authors list them as separate inclusion criteria for the study.

Reply: Thank you for pointing this out, this is corrected to state “signs of conjunctivitis which include chemosis and conjunctival hyperemia” (lines 204-205)

The statement “mild fluorescein stain uptake consistent with mild keratitis” does not make sense. Corneal fluorescein retention indicates absence of the epithelial (with stromal exposure). This can occur with or without corneal inflammation (ie, keratitis).

Reply: This statement has been revised (line 222-223)

Please provide a reference to support the statement “many cats with a primary viral conjunctivitis often have secondary overgrowth of local bacterial flora like Pasteurella multocida”

Reply: A references to this statement (ref 33) was already included. The statement has been reworded to include the reference (lines 210-213).

Fluorescein does not assess for keratitis

Reply: This has been corrected.

Can the ocular disease scoring system be referenced? This is an overly simple system that does not make much sense to this reviewer. Why was it selected when so many published, more robust systems are avaialble?

Reply: This has been reworded and the reference to the scoring system now noted (Line 228-230) 

Why were some cats not sampled for PCR?

- Reply: In some cases, the shelter veterinarians evaluating treated animals failed to collect the required samples, for which the authors apologize

For cats with experimental infections:

-Please provide the specific age of the research cats. Depending on the FHV-1 stain used, clinical disease can be heavily influenced by the cats age.

Reply: Age of research cats (11 mos) now added (line 240)

-Why did the authors use a re-infection model? These seems like a very odd choice and is not representative of a natural scenario. Recurrent disease can be induced in latently infected cats after experimental infection using a variety of mechanisms, or you could simply have evaluated cats with primary infections. How does this system affect the ocular disease and viral shedding patterns that are observed? Please reference this model system that you state was used “to model recrudescent herpesvirus ocular infection”

Reply: Thank you for this comment and question and we agree the primary infection or recurrent infections after glucocorticoid or cyclosporine induced immune suppression models could have been used, but would likely have confounded the evaluation of an immunotherapy. However, in this proof of concept experiment, the recurrent model was chosen both because the cats were available at the time in our colony and by using these cats we could avoid purchasing and inducing disease in additional purpose-bred cats (ie, reduction in animal usage). Because FHV-1 infection does not lead to sterilizing immunity, we expected mild signs to occur after repeat infection, which would provide us with a more realistic model to assess our treatment hypotheses. Appropriate edits to the text are noted (lines 243-247)

-Who were the “Two trained individuals masked to the treatment groups”? Where these clinically trained ophthalmologists (veterinary or human)? Technicians? What examination equipment did they use?

Reply: We have addressed in the text, lines 251-257.

Results

As before, please explain the “trained observer” in detail and how they examined the cats (including what equipment).

Reply: See explanation above

Discussion

“describer” typo (corrected)

Another limitation is that you don’t know you were actually treating cats with FHV-1 in the shelter population (a positive PCR test can be incidental to the ocular disease).

Reviewer #3: The authors have proposed nanoparticle-based treatment for FHV-1 in cats. This model can be a substitute for the rodent models due to the scope in studying FHV-1 reactivation. The authors show that treatment with Liposome dual TLR agonist (LTAC) reduces the disease severity in naturally infected cats and in experimental viral re-infection model. Though the results presented in this work is interesting there are few concerns which needs to be addressed.

1. Fig:2- In the in-vitro study, number of biological replicates should be >3.

Reply: The appropriate number of biological replicates is typically dictated by the sample standard deviation and the size of the treatment effect. In our studies, the treatment effects were large enough, and the sample standard deviation small enough, that we believe the appropriate statistical inferences can be drawn with n = 3 biological replicates. 

2. Stability and half-life of LTAC has not been estimated.

Reply: In unpublished studies, we have shown that the LTAC drops retain full biological potency, as measured by cytokine release assays with cell lines, for at least 3 months when stored at room temperature or at 4C.

3. Fig:3 -Authors show the cytopathic effect seen in Fcwf-4 cells after FHV-1 infection. However, the actual reduction in replicating virus was not calculated. Therefore, the results do not support the authors’ statement “suppression of FHV-1 replication”. FHV-1 replication can be measured from the culture supernatant or from the cell lysate.

Reply: Yes, we agree with the reviewer’s point that reduction in cytopathic activity is not the same as reduction in viral replication demonstrated by virus quantitation, and we have amended the manuscript to reflect this distinction.

4. Fig:6 -The parameters used to calculate the % of inflamed eyes is not clear. Was it based on scoring? Representative eye images can help in understanding the reduction in inflammation.

Reply: This is now addressed on line 274

5. Since FHV-1 can establish latency, the local increase in IFNs response after LTAC treatment can increase FHV-1 latency. The experiments conducted in this work were for 1-2 weeks. Is there data available for recurrence of infection in cats which were naturally infected with FHV-1?

Reply: Thank you for the excellent question. Cats infected naturally with FHV-1 can have recurrence of shedding and clinical illness after stress or immune suppressive events, as is true of HSV infection in humans. This recurrence disease has also been documented in experimental infections in cats as described in reference 26.

---

## [Editor Report · Decision Letter 1]

7 Dec 2022

Nanoparticle Ocular Immunotherapy For Herpesvirus Surface Eye Infections Evaluated in Cat Infection Models

PONE-D-22-23287R1

Dear Dr. Dow:

We’re pleased to inform you that your manuscript has been judged scientifically suitable for publication and will be formally accepted for publication once it meets all outstanding technical requirements.

Kind regards,

Homayon Ghiasi, PhD

Academic Editor

PLOS ONE
---

## [Editor Report · Acceptance letter]

26 Dec 2022

PONE-D-22-23287R1 

Nanoparticle Ocular Immunotherapy For Herpesvirus Surface Eye Infections Evaluated in Cat Infection Models 

Dear Dr. Dow:

I'm pleased to inform you that your manuscript has been deemed suitable for publication in PLOS ONE. Congratulations! Your manuscript is now with our production department. 

Kind regards, 

on behalf of

Dr. Homayon Ghiasi 

Academic Editor

PLOS ONE